# Revisiting "Qualitatively Characterizing Neural Network Optimization Problems"

**Jonathan Frankle**
MIT CSAIL
jfrankle@csail.mit.edu

## Abstract

We revisit and extend the experiments of Goodfellow et al. (2014), who showed that—for then state-of-the-art networks—"the objective function has a simple, approximately convex shape" along the linear path between initialization and the trained weights. We do not find this to be the case for modern networks on CIFAR-10 and ImageNet. Instead, although loss is roughly monotonically non-increasing along this path, it remains high until close to the optimum. In addition, training quickly becomes linearly separated from the optimum by loss barriers. We conclude that, although Goodfellow et al.'s findings describe the "relatively easy to optimize" MNIST setting, behavior is qualitatively different in modern settings.

## 1  Introduction

Neural network loss landscapes are non-convex, yet optimizing with stochastic gradient descent (SGD) leads to optima that achieve low training and test error. Five years ago, Goodfellow et al. (2015) made a set of qualitative observations about neural network loss landscapes in an effort to address this apparent disconnect between the hypothetical difficulty and empirical success of optimizing neural networks with SGD.

For then state-of-the-art networks, Goodfellow et al. linearly interpolated between the state of the network at initialization and the state of the network after training and computed the loss at each point along the this path. They found that loss decreased gradually and monotonically in an "approximately convex" fashion, meaning "that there exists a linear subspace in which neural network training could proceed by descending a single smooth slope with no barriers." Their conclusion was that "the success of SGD on a wide variety of tasks" is because "these tasks are relatively easy to optimize" due to the absence of "complicated obstacles with multiple different low-dimensional subspaces."

Although the nature of neural network optimization has evolved little in the past five years, the networks and tasks considered necessary to support empirical claims have changed substantially. Goodfellow et al. (2015) studied fully-connected and convolutional networks for MNIST that used maxout (Goodfellow et al., 2013) and regularization via both dropout (Srivastava et al., 2014) and adversarial training (Goodfellow et al., 2014). Several of these techniques are no longer in use, and experience shows that many phenomena on MNIST do not occur in general. All of these observations predate VGG-style networks (Simonyan & Zisserman, 2015), ResNets (He et al., 2016), batch normalization (Ioffe & Szegedy, 2015) and, more broadly, the expectation that empirical claims are verified on large-scale settings like ImageNet (Deng et al., 2009).

In this paper, we revisit and extend the observations of Goodfellow et al. (2014) in modern settings. In addition to repeating the experiments on a fully-connected network for MNIST, we examine VGG-16 and ResNet-20 for CIFAR-10 and ResNet-50 for ImageNet. Our findings are as follows:

Workshop on Deep Learning and Information Geometry (NeurIPS 2020), Vancouver, Canada.

- In larger-scale settings, although loss and error are roughly monotonically non-increasing when interpolating along the line between the initial and final points of training, the objective function is not "approximately convex."

- Only on MNIST does it appear that "if we knew the correct direction, a single coarse line search could do a good job of training a neural network." In larger-scale settings, loss plateaus and error remains at the level of random chance along this line segment until near (networks for CIFAR-10) or at (ResNet-50 for ImageNet) the optimum, making it unlikely that an optimization procedure could follow this path.

- In larger-scale settings, the behavior of the linear path to the optimum changes substantially if we interpolate after a small amount of training: as early as iteration 16, the path encounters a barrier. We interpret this to mean that the aforementioned findings may be specific to initialization (where loss is exceptionally high) rather than evidence that "SGD never encounters exotic obstacles." This is not the case for MNIST, which supports the notion that MNIST may be "relatively easy to optimize."

We conclude that, when running the experiment of Goodfellow et al. (2014) in modern settings, (a) the behaviors are sufficiently different that the way Goodfellow et al. interpret their observations no longer applies and (b) the focus on initialization alone ignores important aspects of optimization that complicate the picture painted by Goodfellow et al.

We emphasize that none of our findings are intended as criticism of Goodfellow et al. (2014), who rigorously studied these questions with the best scientific tools available in an era that predated frameworks like TensorFlow and PyTorch, techniques like ResNets and BatchNorm, and bountiful amounts of affordable compute. This paper continues to be well-cited to the present, so we seek to update these observations in the settings that deep learning research focuses on today.

## 2 Preliminaries

We study four image classification settings: a fully-connected network for MNIST with two hidden layers (300 and 100 units), VGG-16 for CIFAR-10, ResNet-20 for CIFAR-10, and ResNet-50 for ImageNet. We use these networks and datasets as implemented in OpenLTH;[1] we use the default hyperparameters and data augmentation from OpenLTH, and all networks reach standard accuracy.

We linearly interpolate using 100 evenly spaced points between the state of the network at iteration $t$ and the state of the network at the end of training. We use $t \in \{0\} \cup \{2^k \mid k \leq 10\} \cup \{1000k \mid k > 1\}$. Unlike Goodfellow et al. (2014), we only evaluate error and loss on the test set due to the computational infeasibility of using the ImageNet training set. We repeat all experiments across three replicates with different random seeds, and we plot the mean and standard deviation.

## 3 Linearly Interpolating from Initialization

In Figure 1, we repeat the main experiment of Goodfellow et al. (2014): examine the loss and error when linearly interpolating from the state of the network at initialization, i.e., before any training has taken place (referred to as *Iteration 0* in Figure 1), to the state of the network at the end of training (referred to as *Last Iteration* in Figure 1).

**MNIST.** On MNIST, we confirm the results of Goodfellow et al. (2014): the loss indeed "has a simple, approximately convex shape along this cross section," decreasing rapidly close to initialization and reaching a "flat region" close to the optimum. Error follows a similar pattern.

**Larger-scale settings.** On the larger-scale settings, loss remains close to its initial value along most (VGG-16 and ResNet-20) or all (ResNet-50) of the line segment between the states of the network at initialization and after training. Loss only becomes convex close to the optimum if at all; for ResNet-50, our resolution of interpolation was not sufficient to detect any decrease in loss prior to reaching the optimum. For ResNet-20 ($x \approx 0.6$) and ResNet-50 ($x \approx 0.99$), loss actually increases slightly. We conclude that, although loss is roughly monotonically non-increasing (with slight deviations on the ResNets), the behavior in larger-scale settings is qualitatively different than the behavior we and Goodfellow et al. (2014) observed for MNIST.

---

[1] https://github.com/facebookresearch/open_lth

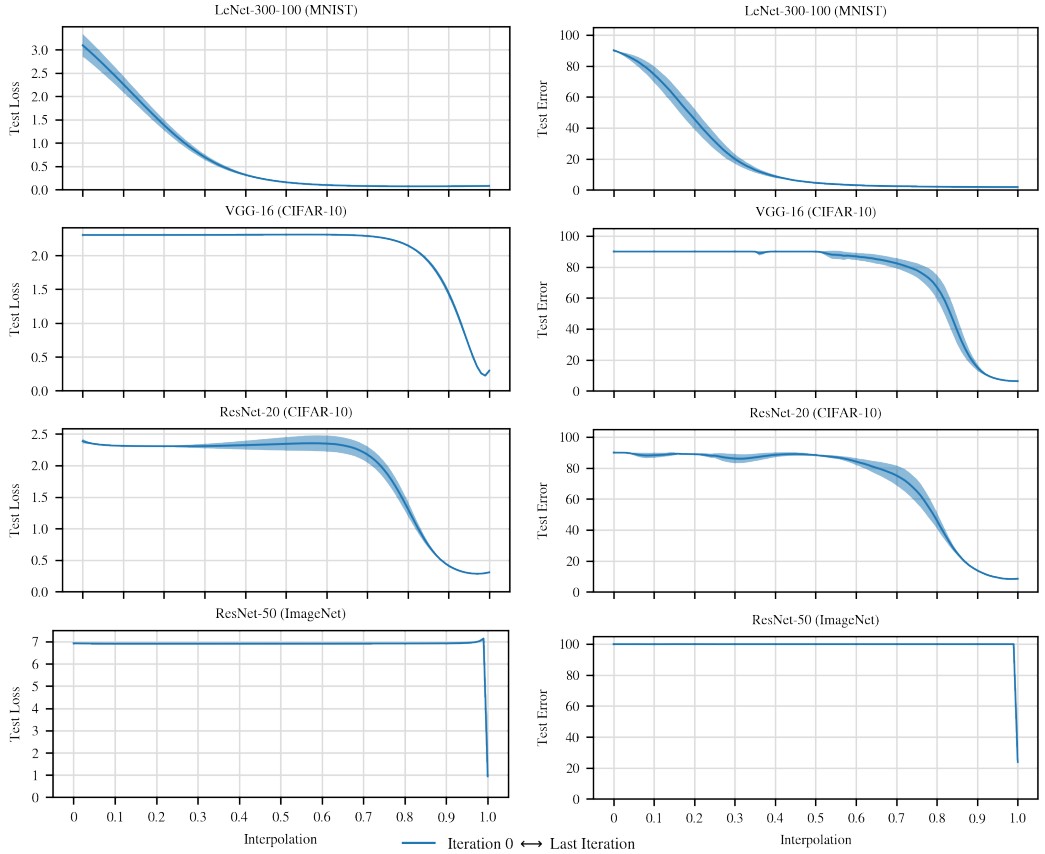

Figure 1: The test loss (left) and error (right) when linearly interpolating at 100 points from the state of the network before training ($x = 0$) to the state of the network after training ($x = 1.0$). Comparable MNIST results are in Figure 4a of Goodfellow et al. (2014).

## 4 Linearly Interpolating from Other Points in Training

In Figure 2, we extend the experiments of Goodfellow et al. (2014) by examining the loss and error when linearly interpolating from the state of the network at other iterations $t$ in training to the state of the network at the end of training.[2]

**MNIST.** On MNIST, the test loss retains its convex behavior when interpolating from later iterations. After iteration 1000, the network is close to its final accuracy, and loss is flat when interpolating.

**Larger-scale settings.** On the larger-scale settings, linearly interpolating after initialization encounters loss barriers. VGG-16 and ResNet-20 encounter increases in loss starting at about iteration 16 (out of 62,560). ResNet-50 encounters barriers starting at about iteration 256 (out of 112,590). These results suggest that, after initialization, the optimum quickly becomes linearly inaccessible from the path that SGD follows. Either SGD makes poor decisions during the early stage of learning that makes the optimization process unnecessarily difficult, or interpolating from initialization is in some sense trivial (due to the especially poor performance of the unoptimized network) and does not reflect full extent of the challenge of optimizing a neural network with SGD.

The optimum becomes linearly accessible again later in training. In Figure 3, we show the maximum increase in loss and error when interpolating from the state of the network at iteration $t$ to the state of the network after training (i.e., the height of the peak moving from left to right in Figure 2). After iteration 8K, 2K, and 50K for VGG-16, ResNet-20, and ResNet-50, there is no longer a barrier.

---

[2]In Appendix B of their paper, Goodfellow et al. (2014) study how far optimization strays from this linear path during training, but they do not interpolate from these points to the end of training.

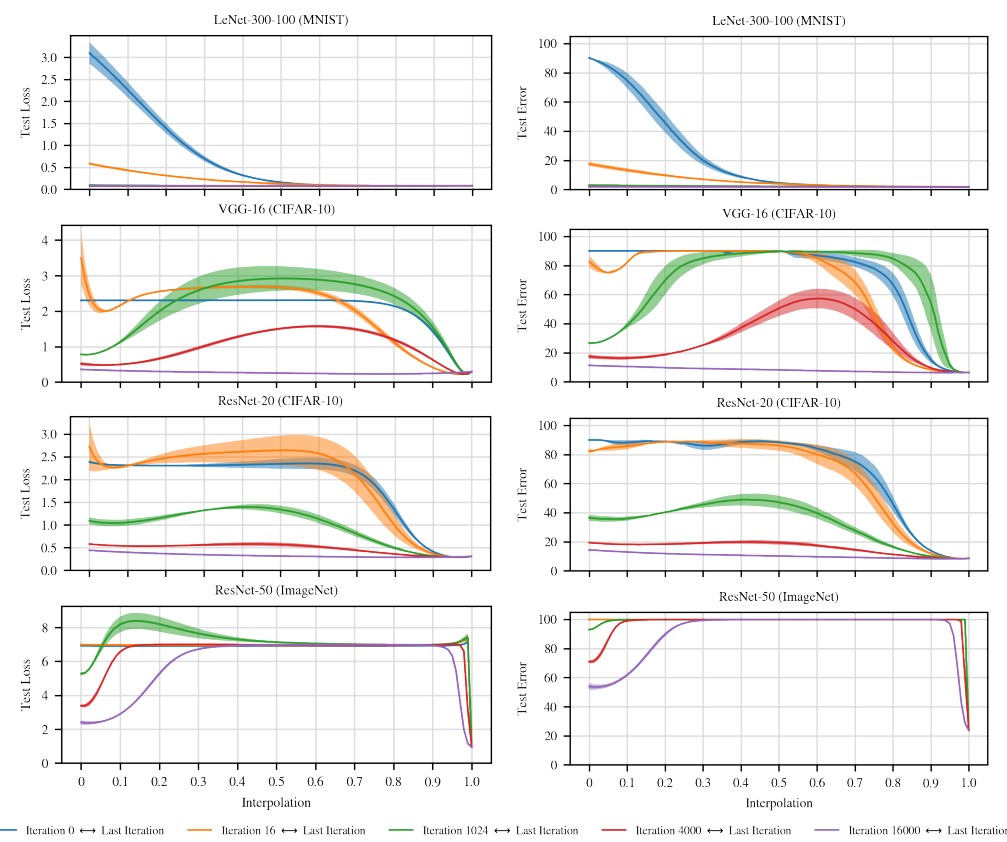

Figure 2: The test loss (left) and error (right) when linearly interpolating at 100 points from the state of the network at the specified iteration ($x = 0$) to the state of the network after training ($x = 1$).

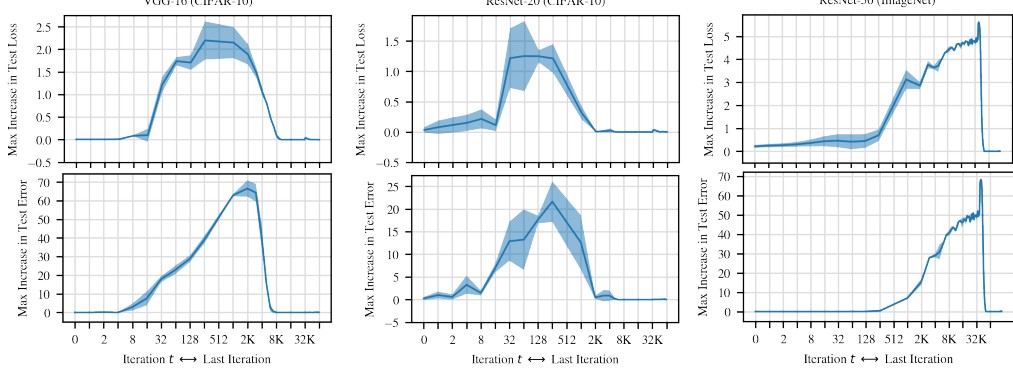

Figure 3: The maximum increase in loss (top) and error (bottom) along the linear path between the state of the network at iteration $t$ (on the x-axis; log scale) and the state of the network after training.

## 5   Discussion and Conclusions

We revisited the observation of Goodfellow et al. (2014) that, for networks trained on MNIST, the loss is a"simple, approximately convex" function along the linear path between initialization and the last iteration of training. We found that, although this is the case on MNIST, the loss remains high until close to the optimum on modern, larger-scale settings. In addition, this property is specific to initialization; soon thereafter, a barrier emerges between the state of the network and the eventual optimum. We conclude that, although we do not find evidence in modern settings for Goodfellow et al.'s interpretation that the loss landscape lacks "complicated obstacles," linear interpolation remains a useful tool for qualitatively characterizing neural network optimization problems.

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
