# OpenReview forum: "Revisiting "Qualitatively Characterizing Neural Network Optimization Problems""
_NeurIPS.cc/2020/Workshop/DL-IG — NeurIPSW 2020: DL-IG Poster_

### Official Review · AnonReviewer2 · 2020-10-22
**Review of "Revisiting "Qualitatively Characterizing Neural Network Optimization Problems""**

**Rating:** 7
**Confidence:** 4

**Review:**

This paper investigates how the loss function behaves on the interpolating path from an initialization of weights to the weights of the final trained network. The main goal is to check whether the observation in Goodfellow et al. -- that the loss looks approximately convex -- continues to hold for larger datasets and more complex neural networks. Through detailed experiments, the authors conclude that the phenomena observed in Goodfellow et al. are limited to the MNIST dataset and do not occur in large datasets with complex models.

I think this is a worthwhile experiment to perform and the authors' results are certainly interesting. Many alternate theories for why SGD performs well have now appeared in the literature, so I would request the author to provide a brief literature review of this topic.

---

### Official Review · AnonReviewer1 · 2020-10-26
**Review of Revisiting "Qualitatively Characterizing Neural Network Optimization Problems"**

**Rating:** 6
**Confidence:** 4

**Review:**

This paper qualitatively explores the loss landscape for neural networks and compares the properties of modern successful networks to architectures that were studied in the past.  The conclusion is that modern architectures do not exhibit convex subspaces between initial and final parameters as previous architectures did.

While this exploration seems interesting, I recommend a few directions to flesh out for future work. First of all, there is a large body of literature, both empirical and theoretical on the behavior of loss landscapes for different types of architectures. I recommend reading about the work on Neural Tangent Kernels which Alemi sometimes calls infinitely wide networks in some of his papers. The reason is that in this limit of very wide networks, it seems that the dynamics of gradient descent becomes approximately linear (Lee et al paper in NeurIPS 2019, for example, but there are many, many papers on this topic). To that end, I also recommend including Wide Resnets in the experiments, since they explore this limit of large width which may have theoretically distinct properties. I also believe that there are papers discussing how residual connections affect the loss landscape, it would be good to use these to inform your discussion.

---

### Decision · Program_Chairs · 2020-11-07

**Decision:**

Accept (Poster)

**Comment:**

Please incorporate the reviewer feedback into your manuscript.